# Which MRI Score and Technique Should Be Used for Assessing Crohn’s Disease Activity?

**DOI:** 10.3390/jcm9061691

**Published:** 2020-06-02

**Authors:** Ferdinando D’Amico, Thomas Chateau, Valérie Laurent, Silvio Danese, Laurent Peyrin-Biroulet

**Affiliations:** 1Department of Biomedical Sciences, Humanitas University, Pieve Emanuele, 20090 Milan, Italy; damico_ferdinando@libero.it; 2Department of Gastroenterology and Inserm NGERE U1256, University Hospital of Nancy, University of Lorraine, 54500 Vandoeuvre-lès-Nancy, France; TChateau@chu-grenoble.fr; 3Department of Hepato-Gastroenterology, University Hospital of Grenoble, University of Grenoble Alpes, 38000 Grenoble, France; 4Department of Radiology, Nancy University Hospital, Lorraine University, 54500 Vandœuvre-lès-Nancy, France; v_croiselaurent@yahoo.fr; 5IBD Center, Department of Gastroenterology, Humanitas Clinical and Research Center -IRCCS-, Rozzano, 20089 Milan, Italy; sdanese@hotmail.com

**Keywords:** MRI score, Crohn’s disease, disease activity

## Abstract

Magnetic resonance (MR) enterography is assuming an increasingly important and central role in the management of patients with Crohn’s disease, as it is not only involved in diagnosis and staging of the disease but also allows for patients’ follow-up, evaluating the response to therapy, and predicting disease outcomes. Several MR scores have been developed but unfortunately there is no globally accepted score. The most widely used scores are the Magnetic Resonance Index of Activity (MaRIA) score, the London score, the Nancy score, and the Clermont score; however, there are multiple differences between these tools in terms of the assessed radiological items, fasting, administration of oral or intravenous contrast, and the use of diffusion-weighted images. Here we reviewed the main characteristics of each MR scoring system to clarify which is the most appropriate score for monitoring disease activity in both clinical practice and clinical trials.

## 1. Introduction

Crohn’s disease (CD) is a chronic, immune-mediated condition with a remitting and relapsing course that can affect any part of the gastrointestinal tract, even if it mainly involves the terminal ileum and colon [1]. CD has a progressive course, and if not adequately controlled, it could lead to several complications over time, such as strictures, fistulas, and abscesses, often requiring surgical treatment [2]. In the last few decades, many biological drugs have been developed and subsequently approved [3] for CD treatment, from anti-TNFα inhibitors (infliximab, adalimumab, and certolizumab pegol), to anti-integrins (vedolizumab) and anti-interleukin 12/23 drugs (ustekinumab); these drugs reduce inflammation and considerably improve the patients’ quality of life [4]. Despite the introduction of these new molecules, a high percentage of patients do not respond to treatments [5]. In this context, the patients’ tight monitoring plays a key role in guiding therapeutic decisions, assessing the response to drugs, and predicting the onset of relapses [6]. A colonoscopy is a fundamental procedure for CD assessment as it not only allows for evaluating endoscopic healing but it is also useful for performing biopsies and analyzing the microscopic disease activity [7]. However, an endoscopic examination requires sedation, is poorly tolerated by patients, does not evaluate the small intestine entirely, and does not provide information on the stenosis type (inflammatory/fibrotic) or possible extra-luminal complications (e.g., fistulas and abscesses) [8,9,10]. These limitations can be overcome through the use of accurate and widely accepted cross-sectional imaging techniques (magnetic resonance imaging (MRI), bowel ultrasound (BUS), and computed tomography (CT) scans) that assess the extent and transmural activity of the disease [11]. Magnetic resonance enterography (MRE) is complementary to endoscopy for the diagnosis, staging, and monitoring of CD patients [12]. MRE takes into account many radiological findings, including segmental mural hyper-enhancement, wall thickening, intramural edema, strictures, ulcerations, restricted diffusion, sacculations, and diminished motility [13]. Many MR scoring systems have been developed to measure disease activity, although there is no standardization on the score to be used [14]. Furthermore, the first instrument for the evaluation of the cumulative bowel damage in CD, namely the Lémann index (LI), has recently been validated [15], but this tool will not be discussed in our work since we are focusing on the assessment of disease activity. This article concisely summarizes the literature data on MRI scores in CD patients, underlining the strengths and limitations of the currently available tools to identify the ideal MR score for monitoring patients in both clinical practice and clinical trials.

## 2. Magnetic Resonance Scores without Diffusion-Weighted Images: The MaRIA and London Scores

The main characteristics of the most-used MR scores are summarized in Table 1. The Magnetic Resonance Index of Activity (MaRIA score) [16] is a partially validated score [17] that assesses wall thickness; post-contrast wall signal intensity; relative contrast enhancement (RCE); and the presence of edema, ulcers, pseudopolyps, and lymph node enlargement. The MaRIA score accurately predicts the response to therapy in CD patients and it is closely related to the Crohn’s Disease Endoscopic Index of Severity (CDEIS) and the Simplified Endoscopic Score for Crohn’s Disease (SES-CD) [18]. A moderate significant concordance (κ = 0.49; *p* < 0.01) was also found between the MaRIA score and the Harvey Bradshaw Index (HBI), revealing that patients with a reduced MaRIA score had a greater chance of clinical improvement based on the HBI (*p* < 0.01) [19]. However, the score is not free from limitations since it is time-consuming and it does not assess the length of disease involvement. Recently Ordás et al. [20] developed a simplified version of the pre-existing MaRIA [17], called simplified MaRIA (MaRIAs), to overcome the practical limitations of the original score [17]. MaRIAs consists of three radiological items from the MaRIA (mural thickening, mural edema, and mucosal ulcerations), and a new item (fat stranding, defined as the loss of the normal sharp interface between the intestinal wall and mesentery, with edema/fluid in the perienteric fat) was added. A strong correlation was found between MaRIAs and CDEIS (*r* = 0.83) and between MaRIAs and the original MaRIA (and *r* = 0.93) (*p* < 0.001). MaRIAs’ interrater agreement among the radiologists was excellent (intraclass correlation coefficient = 0.85; 95% CI: 0.78–0.90; *p* < 0.001) and the calculation time for MaRIAs was significantly shorter compared with the time for standard MaRIA (4.77 min vs. 17.14 min; *p* < 0.001). A recent study [21] that included 121 patients with CD confirmed the linear correlation between MaRIA and MaRIAs (*r* = 0.93; 95% CI: 0.9–0.95). MaRIAs measurements showed a statistically higher interrater agreement compared to MaRIA (96.1% vs. 79.3%; *p* < 0.001) and it required less time on average than MaRIA (4.50 min vs. 12.35 min) [21]. Another relevant tool is the Crohn’s Disease MRI Index (CDMI), also called the “London score.” It is partially validated and it has been specifically structured through the correlation between MRE findings and histology to evaluate non-perforating small bowel CD22. The London score is simple and requires little time for its measurement, but similarly to MaRIA, it does not assess the disease’s extent. A study by Jairath et al. [18] revealed that MaRIA and London scores had a substantial intra-rater and inter-rater reliability (0.89 and 0.71 vs. 0.87 and 0.67, respectively), suggesting a potential role in patients’ assessment for these instruments in both clinical trials and clinical practice.

## 3. Magnetic Resonance Scores with Diffusion-Weighted Images: The Nancy and Clermont Scores

### 3.1. Diffusion-Weighted Imaging Technique

MRE requires the ingestion of large volumes of fluids to distend the small bowel wall and to visualize the lumen [25]. Several oral contrast agents are available for MRE that allow for improving the detection of wall thickenings, fistulas, abscesses, masses, and other mural abnormalities [26,27,28,29]. However, the intake of these preparations is poorly tolerated by patients due to their significant volume (up to two liters of preparation depending on the protocols), their taste, and their possible adverse events (nausea, vomiting, abdominal pain, diarrhea, and rarely, paradoxical constipation) [30]. The introduction of MRI with diffusion-weighted imaging (DWI), a specialized MRI technique, has changed the assessment of CD patients. MRI with DWI does not require the intake of contrast material since it maps the diffusion of water molecules in biological tissues by detecting and grading inflammation and mucosal ulcerations (not for DWI) in small-bowel CD [31]. This technique was initially developed in the neurological field but it is increasingly applied for the study of the abdominal organs, and in particular for the evaluation of patients with inflammatory bowel disease (IBD) [32,33,34]. DWI analyzes the changes in the Brownian motion of fluids following the interactions between cells, macromolecules, and tissues, quantifying the restriction of motion [35]. When there is an increase in cellular concentration in the intra- and extra-cellular spaces and vessels, as in the case of inflammation, fluids’ movement is reduced and the signal diffusion is improved [34]. DWI uses ultrafast spin echo echoplanar T2-weighted sequences and measures the diffusion gradient by calculating the b-value (or diffusion coefficient), which is the strength of the diffusion-sensitizing gradient [36]. In an environment where liquids can flow easily (e.g., blood vessels), the b-value will be low (50–100 s/mm^2^), resulting in a signal loss and providing a black image [36]. On the other hand, in a highly cell-rich environment, the fluids’ motion will be restricted and the b-value will be high (500–1000 s/mm^2^), resulting in a bright image [36]. Interestingly, the qualitative evaluation of the DWI is related to the different intensities of the received signal, while the quantitative assessment of DWI is calculated through the attenuated diffusion coefficient (ADC) and requires at least two b-values (one high and one low), even if multiple values of b can be used, which increases the scanning time [36]. MRI with DWI showed high sensitivity (93%) and acceptable specificity (67%) in identifying the presence of active intestinal inflammation [37], and its main advantage is the lack of need for an intravenous contrast [37]. Although gadolinium is a relatively safe contrast, several studies have shown brain contrast accumulation in both nephropathic patients and those without kidney diseases undergoing repeated resonances [38,39]. A prospective non-inferiority study [37] found no statistically significant difference between MRE with DWI and contrast-material-enhanced (CE) MRE in the inflammation evaluation of CD patients. Moreover, a recent Crohn’s-disease-focused panel of the Society of Abdominal Radiology (SAR) recommended performing cross-sectional imaging in all patients with an initial diagnosis of CD to detect small bowel inflammation and penetrating complications [12]. CT enterography (CTE) and MRE have a similar performance regarding small bowel CD evaluations but MRE should be preferred as it does not expose patients to ionizing radiation [12]. Importantly, if the intravenous contrast cannot be administered, the use of DWI sequences is recommended as a valid alternative [12].

### 3.2. Diffusion-Weighted Imaging Scores

Recently, the use of DWI has been included in two MRE scoring systems: the Nancy and Clermont scores. The Nancy score [40] was the first score developed to qualitatively analyze DWI sequences (hyperintensity or not). It assesses six variables (DWI hyperintensity, rapid gadolinium enhancement, differentiation between the mucosa–submucosa complex and the muscularis propria, bowel wall thickening, edema, and the presence of ulceration) in six intestinal segments (rectum, sigmoid region, left colon, transverse colon, right colon, and ileum). The Nancy score ranges from 0 to 36 and a score >2 is closely related to endoscopic inflammation according to SES-CD (*r* = 0.539, *p* = 0.001) [40] and CDEIS (κ = 0.5, *p* = 0.0015) [23]. Importantly, the Nancy score has recently been validated in an independent cohort of CD patients and accurately predicts mucosal healing and the risk of surgery [23]. A Nancy score <6 showed a good accuracy (sensitivity of 70%, specificity of 80%, positive predictive value of 78%, and negative predictive value of 73%) when identifying mucosal healing using MR before and after treatment initiation, with a low cumulative probability of intestinal surgery (hazard ratio (HR): 1.73, 95% CI: 1.13–2.66, *p* = 0.01) [23]. The operating properties of the Nancy score are broadly similar to those of MaRIA and both scores are highly sensitive to change (Guyatt’s responsiveness index of 1.18 and 1.20, respectively) [20,23,41]. Interestingly, the Nancy score and MaRIAs share many similarities and radiological signs, as three of the four items used in the MaRIAs (wall thickening, parietal edema, and mucosal ulceration) are also found in the Nancy score (Table 2).

The Clermont score (CS) incorporates DWI and the quantitative apparent diffusion coefficient (ADC). The CS obtained external validation in a prospective observational study [42] of 130 subjects that compared MaRIA and CS for the assessment of CD inflammation. The CS was highly correlated with MaRIA in ileal CD (*p* = 0.99) but not in colonic CD (*p* < 0.80). Similar to the original MaRIA score, the assessment is time-consuming and the ADC makes the CS difficult to reproduce since this method is not standardized and there are several post-processing techniques used to acquire the ADC [30]. A retrospective study by Rimola et al. [43] investigated the diagnostic accuracy of three MR scores (MaRIA, CS, and London index). MaRIA showed the best operational characteristics regarding predicting the activity of disease according to SES-CD compared to the Clermont and London scores (0.88, 0.89, and 0.71 for sensitivity, and 0.97, 0.78, and 0.99 for specificity, respectively) (Table 3). The execution of the standard MRE and the use of MaRIA involve fasting, the injection of contrast material, and the intake of a bowel preparation, which is poorly accepted, affecting patients’ compliance and procedure tolerability. The Nancy score could allow for overcoming these limitations, substantially improving the tolerability of the radiological exam. The incorporation of both the Nancy score and MaRIAs in upcoming CD trials will allow for determining the best MR score to use in clinical trials, to identify a cut-off value for MR findings to predict disease activity, and to define the role of MR to clarify whether MR can replace endoscopy for the assessment of CD patients.

### 3.3. Predicting Outcomes with Magnetic Resonance Imaging

A study by Hallé et al. explored the radiological results of 115 patients who had received at least two MREs [44]. After a median delay of 266 days between the two diagnostic procedures, the response to treatment (including thiopurine, methotrexate, adalimumab, infliximab, golimumab, ustekinumab, or vedolizumab) was analyzed [44]. Radiological non-responders (NR) had a higher rate of surgery and endoscopic procedures compared to radiological responders (RR) after 6, 12, and 24 months of follow-up (15% and 7%, 20% and 7%, 32% and 21%, respectively) [44]. Additionally, NR underwent surgery and endoscopic treatments significantly earlier than RR (after 162 and 684 days respectively, *p* = 0.04) [44]. Fernandes et al. [45] confirmed these data, showing that a lower percentage of small bowel CD patients achieving transmural healing required surgery, hospital admission, and therapy escalation (0%, 3%, and 15.2%, respectively) compared to patients with endoscopic mucosal healing (11.5%, 17.3%, and 36.5%, respectively) or active disease (11.6%, 24%, and 54.3%, respectively) after 12 months of follow-up [45]. Interestingly, both the MRE characteristics (proximal bowel dilatation ≥30 mm diameter (OR: 2.98; 95% CI: 1.36–6.55), stricture bowel wall thickness ≥10 mm (OR: 2.42; 95% CI: 1.11–5.27), and stricture length >5 cm (OR: 2.56; 95% CI: 1.21–5.43)) [46], as well as the disease pathways diagnosed using MRE (presence of perianal disease (OR: 9, 95% CI: 2–39, *p* = 0.003), stenoses (OR: 3.4, 95% CI: 1–11, *p* = 0.04), or intra-abdominal fistulas (OR: 10.6, 95% CI: 2–46, *p* = 0.002)) [47], were significantly related with a higher risk of surgery. A study by Deepak et al. [48] investigated the natural history of CD patients managed through a strategy of treating to a target of radiological transmural remission along with a median follow-up of almost 5 years. More than a third of patients per year converted from a response state to a partial/non-response state and 16.7% of patients in a partial/non-response state underwent surgery per year of follow-up [48]. In contrast, no patient in complete remission underwent surgery, suggesting that the achievement and maintenance of radiological transmural healing were associated with surgery avoidance [48]. Importantly, although there is no agreement, several studies have assessed MR scores to identify cut-offs to predict disease activity. A post-hoc analysis [49] of two prospective studies including 63 CD patients showed that bowel wall healing defined according to MaRIA and the Clermont score (no segmental MaRIA > 7 or no segmental Clermont score > 8.4) was associated with a decreased risk of CD-related surgery and a sustained clinical corticosteroid-free remission, including no reappearance or worsening of clinical manifestations leading to therapeutic modification, hospitalization, or surgery. Similarly, in a retrospective single-center study [39], a Nancy score of <6 detected mucosal healing with good accuracy, and radiological remission on MRI with DWI after starting therapy was associated with a lower cumulative risk of intestinal surgery (*p* = 0.0251).

## 4. Conclusions

MRI is acquiring an increasingly central role in the management of CD patients as the treatment target is shifting from the control of symptoms to inflammatory state reduction, including the achievement of radiological transmural healing. Several MR scores are currently available to monitor patients’ disease activity and to evaluate the response to therapy. The recent simplified MaRIA score and the Nancy score share three major items (wall thickening, parietal edema, and mucosal ulceration), are easy to use, and are not time-consuming. These tools have the most suitable characteristics for MRI. Additionally, the use of the DWI technique does not require the administration of bowel preparation, which represents a relevant aspect to be considered. Accordingly, the simplified MaRIA score and the Nancy score should be preferred in clinical practice. Conversely, further comparative studies between the available scores are needed to identify the best score to use in IBD clinical trials and to allow for the homogenization and standardization of patient management.

## Figures and Tables

**Table 1 jcm-09-01691-t001:** Comparison between the most-used magnetic resonance (MR) scoring systems.

Characteristics	MaRIA [16]	London [22]	Nancy [23]	Clermont [24]
Validated score	Yes	Yes	Yes	Yes
Response to therapy	Yes	No	Yes	Yes
Fasting	Yes	Yes	No	Yes
Bowel preparation	Yes	Yes	No	Yes
Radiological Item				
Wall thickness	Yes	Yes	Yes	Yes
Contrast enhancement	Yes	Yes	Yes	No
Edema	Yes	Yes	Yes	Yes
Post-contrast wall signal intensity	Yes	Yes	No	No
Ulcers	Yes	No	Yes	Yes
Pseudopolyps	Yes	No	No	No
Lymph node enlargement	Yes	Yes	No	Yes
Differentiation between M-SM complex and MP	No	No	Yes	No
Comb sign	No	Yes	No	No
Fistulas and abscesses	No	No	No	Yes
Length of disease	No	No	No	No
DWI	No	No	Yes	Yes
Apparent diffusion coefficient	No	No	No	Yes
Correlation with CDEIS	Yes	Yes	Yes	Yes

MaRIA: Magnetic Resonance Index of Activity; DWI: diffusion-weighted imaging; CDEIS: Crohn’s Disease Endoscopic Index of Severity; M-SM: mucosa–submucosa; MP: muscularis propria.

**Table 2 jcm-09-01691-t002:** Comparison between the Nancy score and the simplified MaRIA score.

Characteristics	Nancy Score [39]	MaRIAs [20]
Fasting	No	Yes
Bowel preparation	No	Yes
Intravenous contrast injection	Yes	Yes
Radiological items	Mural thickening Mural edema Mucosal ulcerations	
	DWI hyperintensity Rapid gadolinium enhancement Differentiation between the M-SM complex and the MP	Fat stranding
Sensitivity to change	Guyatt’s responsiveness index: 1.18	Guyatt’s responsiveness index: 1.13
Inter-rater agreement	k coefficient = 0.85	k coefficient = 0.85
Intra-rater agreement	0.96	0.69
Calculation time	4–5 min	4.77 min

MaRIAs: simplified Magnetic Resonance Index of Activity; DWI: diffusion-weighted imaging; M-SM: mucosa–submucosa; MP: muscularis propria.

**Table 3 jcm-09-01691-t003:** Operational characteristics and the main limitations of MRI scores.

Score [Reference]	Diagnostic Accuracy at Identifying Segments with Active CD	Correlation with Endoscopy	Limitations
MaRIA [16,20]	MaRIA > 1 SE: 90% SP: 81% AUC: 0.91	CDEIS (*r* = 0.78) SES-CD (*r* = 0.808)	Intravenous contrast injection Complex score Time-consuming The length of disease involvement is not evaluated The DWI is not evaluated
MARIAs [20]	MARIAs ≥ 2 SE: 85% SP: 92% AUC: 0.94	CDEIS (*r* = 0.83)	Intravenous contrast injection The DWI is not evaluated
London [22]	London > 4.1 SE: 81% SP: 70% AUC: 0.76	SES-CD (*r* = 0.80)	Complex score The length of disease involvement is not evaluated The DWI is not evaluated
Nancy [38]	Nancy < 6 SE: 70% SP: 80% AUC: 0.82	CDEIS (κ = 0.5) SES-CD (*r* = 0.539)	Intravenous contrast injection is needed
Clermont [43]	Clermont > 8.4 SE: 54.4% SP: 81.3% AUC: 0.68	CDEIS (*r* = 0.48) SES-CD (*r* = 0.44)	Time-consuming Complex score ADC is not a standardized method

SE: sensitivity; SP: specificity; AUC: area under the curve; CD: Crohn’s disease; CDEIS: Crohn’s Disease Endoscopic Index of Severity; DWI: diffusion-weighted imaging; SES-CD: Simplified Endoscopic Score for Crohn’s Disease; ADC: apparent diffusion coefficient.

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
