# Peer review of "Which MRI Score and Technique Should Be Used for Assessing Crohn’s Disease Activity?"

_jcm, 2020, doi:10.3390/jcm9061691_

Round 1

Reviewer 1 Report

D'Amico and colleagues, reviewed the literature addressing a relevant topic in the management of patients with Crohns' disease. As the authors reported, MRE is rapidly gaining a central role in clinical decision making and several scores have been proposed. The present manuscript, accurately describes and compares the different MR scores underlining their strengths and limitations in a concise manner.

Author Response

Referee 1

D'Amico and colleagues, reviewed the literature addressing a relevant topic in the management of patients with Crohns' disease. As the authors reported, MRE is rapidly gaining a central role in clinical decision making and several scores have been proposed. The present manuscript, accurately describes and compares the different MR scores underlining their strengths and limitations in a concise manner.

Reply: We gratefully thank the reviewer for his/her positive comment.

Reviewer 2 Report

Dear Authors, 

this manuscript is needed for comprehensive summary of the characterisitc of exisiting MR scores for CD assessment. I have several remarks that will make the review more readable for professionals. 

A. You may use more precize data from cited studies that will help to describe scores in the main body of the text. 

B. The role of MR in the context of endoscopy replacement for CD patients assessment is discussed and further studies will allow to ansewer the question. You review may show detailed characterisitcs of mentioned scores with comparison to endoscopy.  I would suggest to introduce more details in tables about scores and these tables that are already presented should be viewed after the paragraph where are discussed instead the end of the manuscript. 

  1. Please introduce table where you show the sensitivity and specificity each of the discussed scores
  2. Please state main limitations each of the presented scores in separate table
  3. Please show the comparision of the discussed scores with endoscopy - advantage and disadvantage in practical use

C. Try to be more objective in your review and guide professionals when exactly which method is the best for CD assessment. Pehaps then, you can state conclusions based on the present literature and instead of refering to upcomming trial.

Author Response

Referee 2

Dear Authors,

this manuscript is needed for comprehensive summary of the characterisitc of exisiting MR scores for CD assessment. I have several remarks that will make the review more readable for professionals.

Reply: We thank the reviewer for the comment.

  1. You may use more precize data from cited studies that will help to describe scores in the main body of the text.

Reply: We gratefully thank the reviewer for the comment. As requested, we have added specific data from the mentioned studies to better describe scores’ characteristics.

  1. The role of MR in the context of endoscopy replacement for CD patients assessment is discussed and further studies will allow to answer the question. You review may show detailed characterisitcs of mentioned scores with comparison to endoscopy. I would suggest to introduce more details in tables about scores and these tables that are already presented should be viewed after the paragraph where are discussed instead the end of the manuscript.

Reply: Valid suggestion. We have made the appropriate changes as requested.

1.Please introduce table where you show the sensitivity and specificity each of the discussed scores

Reply: We thank the reviewer for the comment. We have added Table 3, including sensitivity and specificity for each score.

2.Please state main limitations each of the presented scores in separate table

Reply: Table 3 includes also the main limitations of each score.

3.Please show the comparison of the discussed scores with endoscopy - advantage and disadvantage in practical use

Reply: We thank the reviewer for the comment. We have included in Table 3 the correlation between MRI scores and endoscopic scores to facilitate data interpretation.

  1. Try to be more objective in your review and guide professionals when exactly which method is the best for CD assessment. Pehaps then, you can state conclusions based on the present literature and instead of refering to upcoming trial.

Reply: We thank the reviewer for his/her comment. The data reported in our review come from literature evidence. Therefore, we have modified our conclusion by providing indications on which score to choose in clinical practice as required.

Round 2

Reviewer 2 Report

Authors provided needed details to improve the manuscript and make it more readible for readers. I recommend for publication in this form.